# Genetic Perception Versus Nutritional Factors: Analyzing the Indigenous Baduy Community’s Understanding of Stunting as a Health Issue

**DOI:** 10.3390/ijerph22020145

**Published:** 2025-01-22

**Authors:** Liza Diniarizky Putri, Herlina Agustin, Iriana Bakti, Jenny Ratna Suminar

**Affiliations:** Faculty of Communication Sciences, Universitas Padjadjaran, Bandung 45363, Indonesia; h.agustin@unpad.ac.id (H.A.); iriana.bakti@unpad.ac.id (I.B.); jenny.suminar@unpad.ac.id (J.R.S.)

**Keywords:** stunting, public health, Baduy community, indigenous healthcare, cultural communication, NGO intervention, healthcare equity

## Abstract

This study investigates the challenges and opportunities in addressing public health issues in the context of stunting in the Baduy community. Baduy is a remote indigenous group in Indonesia. The Indonesian government and NGOs such as SRI and Dompet Dhuafa have attempted to abolish stunting. However, factors such as cultural aspects, communication gaps, and logistic problems prevent the optimization of health interventions. Midwives and other health workers have yet to win the community’s trust and provide quality services, but the lack of sustainable solutions further worsens their problem. This studyhighlights the urgency of culturally appropriate, long-term strategies that stay within the unique Baduy lifestyle and belief system, including integrating the tribal leaders into health campaigns. This study also seeks to explain the role of modern healthcare in the Baduy community, particularly the functional acceptance of modern medicine due to its effectiveness in treating severe health problems. However, controversies regarding access to healthcare for Indigenous peoples, especially regarding government resources for care in urban centers, reveal broader issues of healthcare equity in Indonesia. The study finds the need to advocate improved and culturally sensitive interventions, particularly in health communication and government support, to ensure sustainable improvements in public health for Indigenous peoples such as the Baduy.

## 1. Introduction

Stunting, a form of chronic malnutrition that affects the physical and cognitive development of children, is a major global health issue. According to the World Health Organization (WHO), approximately 22% of children under five years old globally are stunted, with the highest prevalence in low- and middle-income countries. In the short term, stunting increases the incidence of pain and the risk of death; decreases children’s cognitive, motor, and verbal potential; and increases health risks and the lack of learning capacity while in school [1]. Stunting results in a short stature and impairs brain development, leading to lower cognitive abilities, poor school performance, and decreased economic productivity in adulthood. In the long term, stunting has an impact on reducing productivity levels in adulthood and, therefore, lowering income. It contributes to an increased risk of chronic diseases later in life, exacerbating cycles of poverty and ill health across generations [2].

Stunting refers to impaired growth and development in children, primarily due to chronic malnutrition during the first 1000 days of life, from conception to age two. It is measured by low height-for-age and indicates a failure to achieve expected physical and cognitive milestones. The causes of stunting include inadequate nutrition, frequent infections, poor maternal health, and insufficient access to quality food and healthcare. According to the World Health Organization (WHO), globally, approximately 22% of children under five years are stunted. In Indonesia, it is exceptionally high among Indigenous communities, such as the Baduy [3]. The consequences of stunting are severe, including impaired cognitive development, increased susceptibility to diseases, and reduced future earning potential, as highlighted by [4], who estimates that stunting could reduce individual earnings by up to 7%.

The issue of stunting is an urgent matter in Indonesia. The recent data from the Ministry of Health shows that 24% of Indonesian children under five years old suffer from stunting. Such prevalence places Indonesia among the countries with severe public health concerns regarding childhood nutrition. The Indonesian government has launched many programs to solve this issue, including the National Strategy to Accelerate Stunting Prevention (2018–2024). The exerted effort, however, still has yet to eliminate the threat of stunting, especially in vulnerable areas and those with traditional solid cultures and limited access to health services, such as the Baduy community.

The Baduy, an Indigenous ethnic group living in Banten Province, Indonesia, has maintained a distinct cultural identity for centuries. Divided into the Baduy Dalam (Inner Baduy) and Baduy Luar (Outer Baduy), they are known for their compliance toward ancestral traditions, strict environmental conservation practices, and minimal interaction with modern society. The Baduy Dalam, in particular, live in isolation and reject modern technologies, including healthcare and education services, which significantly influence their health and nutritional outcomes [5].

Table 1 illustrates the progress in monitoring the nutritional status of infants and toddlers in the Baduy community over two years, 2022 and 2023. It includes data on the number of infants and toddlers whose nutritional status was tracked and those classified as undernourished (weight-for-age) and stunted (height-for-age). It also highlights the number of cases showing improvement in dietary deficiencies and stunting conditions.

Table 1 shows that while more children were monitored in 2023, the sharp rise in undernourished and stunted cases suggests that the overall nutritional situation in the Baduy community may have worsened or that better identification efforts revealed more cases. Despite interventions, the number of children showing improvement remains small, indicating that existing programs may need to be strengthened or adjusted to achieve more effective and sustainable results.

Cultural beliefs and practices in the Baduy community profoundly shape their understanding of health and disease. Traditional knowledge and customs are passed down through generations, often prioritizing spiritual and ancestral explanations over scientific ones. As a result, health issues such as stunting may be misunderstood or attributed to non-medical causes, such as genetic inheritance or ancestral curses. These cultural perceptions create barriers to introducing modern health interventions and improving public health outcomes in the community. Understanding local perceptions of health is crucial in designing and implementing effective public health interventions. Indigenous communities like the Baduy hold strong cultural beliefs influencing their health behaviors, including nutrition and child-rearing practices [6]. Misconceptions about stunting, particularly the belief that it is mainly a genetic issue, can dampen the effort to alleviate the situation.

Culture affects how communities understand and navigate their way around the world. Indigenous communities such as Baduy often have distinct health perspectives that grow from traditions and culture. These perceptions can significantly influence the community’s understanding of stunting and its causes. Studies [7,8] demonstrate that health literacy is closely tied to cultural norms in many Indigenous populations. The study found that early marriage, a common practice in some cultures, contributes to stunting, as young mothers may lack the knowledge and resources to provide adequate nutrition for their children. Similarly, ref. [9] highlights the importance of involving cultural leaders in health education to improve community awareness and intervention efforts.

There is an ongoing debate about the relative importance of genetic versus nutritional factors in stunting. Like other Indigenous populations, some members of the Baduy community believe that stunting is primarily genetic, which can create a barrier to effective nutritional interventions [10]. Meanwhile, the research found that while genetics may affect the emergence of stunting, poor nutrition and environment remain the determining factors [11]. Several non-biological aspects, including socioeconomic and cultural practices (SEPE), also contribute to stunting [12]. At the same time, studies in different contexts, such as the work of [13], demonstrate that communication for behavior change, combined with the effort to provide nutritional support, may be effective in reducing stunting, which proves environmental—rather than genetic—causes of the condition.

Addressing stunting in communities with strong cultural beliefs, such as the Baduy, has its challenges. The Baduy community’s total conformity to ancient traditions and culture creates significant barriers to introducing modern healthcare interventions. For example, the Inner Baduy Tribe strictly avoids interaction with the outside of their community, including modern medicine, as part of their commitment to preserving their cultural identity. This isolation challenges providing information or services that can alleviate public health problems. The Inner Baduy Tribe is also known to reject health interventions not in line with the tribe’s cultural values. In several cases where health service providers or NGOs have tried to provide stunting prevention counseling, the Baduy Tribe has rejected the efforts and considered it incompatible with their way of life. This condition arises from the desire to protect cultural traditions and low trust in foreigners who bring modernity that conflicts with their spiritual and naturalistic worldview. For instance, concepts of balanced nutrition, child growth monitoring, and food supplementation may be perceived as intrusive or unnecessary, especially if they clash with their belief in self-sufficiency and natural living.

One of the significant barriers to addressing stunting in the Baduy community is the widespread misconception that stunting is primarily a genetic issue rather than a result of malnutrition or environmental factors. This belief is rooted in traditional interpretations of health, where physical traits such as short stature are often seen as predetermined by ancestral lineage and not influenced by diet or living conditions. The Baduy attribute a large part of a child’s physical development to their heritage, with minimal emphasis placed on the role of nutrition in growth. As a result, parents have yet to recognize the importance of proper child-feeding practices or the risks associated with poor nutrition during early childhood development. This misconception has significant implications for public health interventions. When stunting is viewed as unavoidable, the community is less likely to follow preventive efforts such as improving their diet quality, accessing healthcare services, or engaging in maternal and child nutrition programs. The belief in the hereditary nature of stunting reduces the perceived urgency to address the problem, as families may have resigned themselves to the misconception.

Thus, efforts to combat stunting in the Baduy community may only succeed by correcting these misconceptions. Public health programs must go beyond merely providing nutritional support; they must also focus on reshaping the community’s understanding of stunting as a preventable condition tied to dietary factors. Only by addressing the cultural misconceptions surrounding the causes of stunting can effective interventions be developed and implemented in a way that resonates with the Baduy and their unique worldview.

This study is significant in understanding the local perceptions of stunting within the Baduy community, offering insights into the cultural and social factors that shape these views. By examining how Baduy perceives stunting—a genetic condition or a preventable nutritional issue—this studycontributes to the broader effort to approach stunting prevention in culturally sensitive ways. The study aims to answer the following questions: How does the Baduy community perceive stunting? What factors influence their understanding of stunting as a genetic or nutritional problem? The objectives of this study are to explore the perceptions of stunting within the Baduy community and to analyze the implications of these perceptions on public health interventions. The findings will inform strategies that respect the community’s cultural context while addressing the critical nutritional needs of children, bridging the gap between traditional beliefs and modern health interventions to ensure that stunting reduction efforts are effective and culturally appropriate.

## 2. Materials and Methods

### 2.1. Research Design

This study employed a qualitative case study approach, which allowed for an in-depth exploration of the Baduy community’s perceptions of stunting. A case study wasa qualitative approach, in which the researcher exploreda closed system (referred to as a case) or a number of closed systems (cases) over a period of time through in-depth and detailed data collection, involving multiple sources of information (e.g., observations, interviews, audiovisual materials, and documents and reports), and reporteda description of the case and case-based themes [14].

The case study method wasparticularly appropriate for understanding the complex interplay between cultural beliefs, health perceptions, and nutritional practices within a specific, culturally unique group. Qualitative methods provided the flexibility to explore the subjective and socially constructed meanings of health and stunting, which are deeply rooted in the cultural norms and traditions of the Baduy. This approach wasvaluable in understanding how the Baduy community views stunting and how these views influence the success or failure of health interventions targeting stunting.

Qualitative research also allowedfor the researcher to capture the nuances of communication, behavior, and cultural practices that may not be evident in quantitative research. Focusing on the lived experiences and narratives of the community members facilitated a more comprehensive understanding of stunting as it is perceived in a cultural context, providing insights that can inform culturally sensitive health interventions. The Ethics Commission of Padjajaran University, number 40/UN6, KEP/EC/2024, recommended ethical clearance approval.

In this study, the researcher’s position wasas an outsider positionality [15]. Researchers can be said to be outsiders because they have cultural differences [16] with the Baduy community. Researchers are academics who live in an urban environment with a modern culture and academic mindset, while the Baduy community lives in a rural environment with its own culture. As outsiders, we are less likely to form biases because they are not embedded in the social or cultural context of the group being studied; have emotional distance that reduces the likelihood of personal involvement influencing the research outcomes; bring a fresh, external viewpoint that might uncover nuances or issues that insiders might overlook due to familiarity; can focus on evidence-based conclusions rather than relying on ingrained beliefs or norms; and are able to critically analyze the context and question practices or beliefs that insiders might take for granted [17]. However, this position restricted our ability to fully access the complex social networks, cultural subtleties, and localized knowledge [18] that define the Baduy community. However, this status as an outsider wasnot a fixed entity [19]. Over time, we became more accepted by the Baduy community and more involved in their simple cultural context.

### 2.2. Study Area and Population

The study was conducted in the Baduy community, a remote Indigenous group located in the mountainous regions of Banten, Indonesia (Figure 1). The Baduy are divided into two subgroups: the Baduy Dalam (Inner Baduy), who live in isolation from modern society and strictly adhere to traditional customs, and the Baduy Luar (Outer Baduy), who are more open to external influences but still maintain many conventional practices. The study focusedon both subgroups to better understand how traditional versus slightly more modernized cultural contexts influence perceptions of stunting.

The Baduy community is known for its adherence to ancestral practices, particularly in farming, healthcare, and child-rearing areas. Their lifestyles are characterized by minimal access to modern health services, caused by their geographic isolation and cultural norms that discourage dependence on external intervention. Such factors have promoted stunting to the rank of a major problem in the community, made worse by the minimal knowledge of nutrition and beliefs that associate stunting with genetic or ancestral factors rather than a preventable medical condition.

### 2.3. Data Collection Methods

Various data collection methods were employed to fully understand the perceptions of stunting within the Baduy community, ensuring that different perspectives from community members and health workers were considered. The data sources in case-study research are people who have direct knowledge related to the events being studied [14]. In-depth interviews with Baduy community members, two mothers of stunted children, were conducted to explore their beliefs about the causes of stunting, their knowledge of nutrition, and their reliance on traditional health practices. These interviews provided personal narratives highlighting cultural beliefs’ role in shaping health perceptions and behaviors. The interview also sought to uncover any awareness or understanding of stunting as a preventable health issue or whether it was predominantly viewed through a genetic lens. We also conducted interview with village secretary to uncover the steps that have been taken by the village government to increase community awareness on stunting and other health issues.

For outsider perspectives, interviews were held with a local health worker and representatives from NGO involved in health and nutrition programs in the region. The interview aimed to explore the challenges faced by healthcare providers in implementing stunting prevention programs and gauge their perspectives on the community’s responses to these interventions. Local health workers, often positioned between modern healthcare systems and traditional community beliefs, offered critical insights into past and current interventions’ effectiveness and perceived barriers to changing nutritional practices.

Participant observation in the community was employed to gain first-hand daily life experience in the Baduy. Observation allows researchers to see and observe for themselves and record behavior and events as they actually are, complementing interview data collection so that there is no bias or error, and it can collect data in certain cases where data cannot be collected in other ways [21]. By engaging in community activities, the researcher observed dietary practices, parenting behaviors, and community rituals, all contributing to the local understanding of child health and development. This method allowed for collecting rich, contextual data on how stunting is embedded in the community’s way of life, offering insights that might not have surfaced through interviews alone.

In case studies, triangulation of data with multiple sources can increase the validity of research findings [22,23]. Therefore, in addition to interviews with insider and outsider perspectives and participant observation, researchers also used survey results reported by [24], which were conducted at the same time as the research conducted by the researcher.

### 2.4. Data Analysis

The data analysis involved a systematic approach to identify key themes and patterns related to the community’s perceptions of stunting and the role of cultural beliefs in shaping these perceptions. Thematic analysis was applied to the transcripts from in-depth interviews and focus group discussions. It involved coding the data into various themes, using six steps: (1) becomingacquainted with the data, (2) creating preliminary codes, (3) identifying potential themes, (4) refining and validating themes, (5) defining and labeling themes, and (6) compiling the final report [25]. We used a deductive approach, guided by the interview topics, and a semantic approach, focusing on the explicit meanings within the data, to develop our initial codes [25].

## 3. Results

### 3.1. Perception of the Baduy Community Towards Stunting

#### 3.1.1. Community Attitudes Toward Stunting

The Baduy community members interviewed by researchers regarding the stunting conditions affecting their children exhibited a neutral attitude toward the issue. Rather than perceiving stunting as a critical concern, they focused more on the overall health and well-being of the child, prioritizing holistic health over specific risk factors. When a mother was asked if she felt concerned about her son, Andi, who was noticeably smaller than Rambu, a child of the same age but bigger in stature, her response reflected this mindset. She expressed no particular worry, underscoring the community’s perspective that physical size alone does not necessarily indicate a problem as long as the child appears healthy in other respects. This outlook reveals a deeper cultural understanding of health, where individual variation is accepted and external comparisons are downplayed.

Researcher:
*“That is, the child is taller than this child. Are you worried about why this child is different from that child?”*


Mrs. Andi:
*“No, as long as both are safe. It is not because you’re small and worried; as long as it is healthy, it is important to be healthy”.*


Another respondent, a 17-year-old mother with one child, was unfamiliar with stunting. Despite this lack of understanding, she occasionally expressed curiosity about her child’s more diminutive stature than a neighbor’s child, who appeared larger. However, these discussions were brief and needed more subsequent action or follow-up. The mother reported not being concerned about her child’s shorter height, as she did not perceive it as a health issue since it was not accompanied by pain. When asked about her specific concerns, she responded as follows:

Researcher:
*“If I told you that your daughter Sarma is short, would you be worried?”*


Mrs. Sarah:
*“No, it’s normal”.*


Researcher:
*“What worries you?”*


Midwife Ira:
*“For example, if you see your child, what condition makes you worry about her?”*


Mrs. Sarah:
*“If he is cheerful, there is nothing to worry about”.*


Midwife Ira:
*“Maybe crying?”*


Mrs. Sarah:
*“Crying all the time”.*


The respondent’s response indicates that she is not concerned about stunting itself but is more worried about her child falling ill. It suggests that the mother does not perceive a connection between stunting and illness, focusing on more immediate health concerns. Her understanding appears limited to proximal factors that directly impact health. At the same time, she does not consider distal factors, such as stunting, which require a broader understanding of the link to overall health outcomes.

#### 3.1.2. Limited Awareness and Knowledge Gaps

The informants interviewed needed to familiarize themselves with the concept of stunting. When asked about stunting and its relation to child health, they openly admitted not knowing the term and did not have a specific name for the condition. However, they described children who fail to gain weight or grow appropriately as those who “do not want to grow up”, indicating an awareness of growth issues despite the absence of medical terminology; it suggests that while the community recognizes growth-related problems, these are framed within a local, culturally rooted understanding rather than through formal medical concepts, such as stunting. This gap highlights the importance of culturally sensitive health education when addressing issues like stunting in such communities.

Researcher:
*“Midwife Ira has explained stunting, diarrhoea, and health. Do you know what stunting is?”*


Mrs. Andi:
*“No”*


Interviewer:
*“So you do not know about stunting?”*


Mrs. Andi:
*“I do not know”.*


On the community level, the survey in Figure 2 reveals significant gaps in knowledge and healthcare practices related to stunting. A staggering 90% of respondents were unaware of their child’s stunting, and 80% were unfamiliar with the signs of stunting. This lack of awareness may stem from the need for formal education on stunting (87%) and nutrition (67%). Instead, the community heavily relies on traditional healers, with 91% of mothers consulting a medicine woman (paraji), while only 8% sought advice from healthcare professionals. This reliance extends to breastfeeding consultations, with 79% turning to the paraji versus 21% who sought medical advice.

#### 3.1.3. Reliance on Traditional Health Practice

Furthermore, 53% of pregnant women were examined by a paraji during pregnancy, and 56% visited a community health center (posyandu/puskesmas). Traditional medicine is often the first recourse for most women, reflecting the community’s trust in conventional practices.

#### 3.1.4. Cultural Practices and Their Impact on Health

Additionally, cultural practices persist, with 90% of pregnant women working in the fields and 84% bringing their children to the fields, which may affect their health outcomes. These data highlight the need for improved education and the integration of formal healthcare to address stunting in the Baduy community, where traditional beliefs may override awareness of modern health issues like stunting.

#### 3.1.5. Impact of Improved Healthcare Access

The informant acknowledged that the current public health conditions have improved significantly compared to the past, primarily due to increased access to formal healthcare services. Earlier, they relied solely on traditional remedies, such as herbal medicine, to address health concerns. However, with the availability of clinics, midwives, and other healthcare facilities, they now have access to modern treatments. This shift has led to a noticeable improvement in the overall health of their community, highlighting the positive impact of more accessible and reliable health services on traditionally underserved populations.

Researcher:
*“But according to you, how is public health here compared to the past? Is it the same, or has there been a change?”*


Mrs. Andi:
*“It is the same. It is just a mending. Those who do not recover can be helped by treatment. If you want medical treatment now, stay here [do not have to go out of the village]”.*


#### 3.1.6. Perception of Health as Genetic

The mother attributed her child’s small stature primarily to genetic factors, noting that her son’s height is similar to that of the father, who is also of short stature.

Researcher:
*“How old is Sardin?”*


Mrs. Sarah (Sardin’s mother):
*“10 years old”*


Midwife Ira:
*“Try to compare Sardin with his brother. His body size is very far, right, mom?”*


Mrs. Sarah:
*“There’s no problem, his brother is fat. Sardin is small. It’s genetic because his father is also small”.*


#### 3.1.7. Misalignment Between Health Concerns and Interventions

The respondent had previously received a stunting poster and an explanation from a midwife regarding its contents. Despite this, she could not recall the information later, suggesting that the poster could have more effectively engaged her or prompted her to consider its recommendations. It indicates that the information provided did not leave a lasting impact or influence her behavior, highlighting a gap in the effectiveness of the educational materials in fostering meaningful understanding or change.

Midwife Ira:
*“The picture posters on display, explained by Midwife Ira, do you understand what they mean? Do you guys know? Or don’t understand?”*


Mrs. Sarah:
*“I was told, but I forgot”.*


In an interview regarding stunting within the Baduy community, Ambu Sarah exhibited a passive attitude and limited understanding of the condition. When asked about her familiarity with the term “stunting”, her response, “do not know”, indicated a significant lack of awareness. Furthermore, when prompted to describe children who appeared underweight or failed to gain weight, she referred to them as “kuru kitu” (thin) but provided no elaboration or expressed notable concern. This response suggests that such conditions are perceived as usual rather than signs of a health issue requiring intervention.

#### 3.1.8. Proactive Efforts by Village Authorities

The Village Secretary of Kanekes described the local government’s proactive measures to address stunting. The village authorities have implemented strategies to educate and engage the community, disseminating information sourced from health authorities. These initiatives include organizing educational sessions and socialization activities during Posyandu (integrated health service post) meetings, where nutrition and health education are emphasized. Additionally, the village government collaborates with the local Puskesmas (community health center) to enhance outreach and public awareness efforts, striving for a comprehensive approach to tackling stunting.

“*We in the Village have a community health centre that works with volunteers from the province or district or Social Services, who provide a lot of socialization about the impact and dangers of stunting. However, they don’t understand Stunting because Stunting has no sick symptoms or direct evidence. They only participate in activities but do not believe*”. (Village Secretary, 2024)

#### 3.1.9. Persistent Challenges in Community Perception

Despite these efforts, a lack of awareness and understanding about stunting persists. Many within the community view stunting as a hereditaryrather than a preventable health concern. This misconception undermines intervention efforts and underscores the need for sustained education.

“*Stunting in Kanekes is arguably like a ghost. The government says stunting* [is a preventable health concern], *people here say it is hereditary*”.(Village Secretary, 2024)

The secretary noted that the Baduy community typically does not perceive stunting as a medical condition, dismissing it as an inherited trait. This skepticism mirrors the community’s response to other health issues, such as the COVID-19 pandemic, which was also met with significant doubt. Consequently, individuals affected by stunting view it not as a problem requiring concern or intervention.

#### 3.1.10. Barriers to Effective Engagement

This widespread indifference has left village authorities frustrated and overwhelmed. The village secretary attributes this skepticism to the community’s limited access to formal education and the absence of tangible evidence linking stunting to adverse outcomes. In addition, geographical challenges, such as poor signal reception in Baduy Dalam (Western Baduy region), further hinder efforts to deliver health education and services effectively.

### 3.2. Perception of the Non-Governmental Organizations Towards Stunting

#### 3.2.1. NGO Dompet Dhuafa

##### DompetDhuafa’s Philanthropic Mission in Kanekes Village

Dompet Dhuafa is a philanthropic organization that channels community financial support to marginalized groups, including the Baduy community. As part of its mission, the organization has made significant contributions to addressing health and nutrition challenges in the region. One such initiative is the deployment of Midwife Ira, who contracted for two years to serve Kanekes Village. Initially positioned at the health center, she was reassigned to the NGO Sahabat Relawan Indonesia (SRI) due to the health center’s oversaturation of midwives.

##### Midwife Ira’s Role and Reassignment to Address Health Needs

In her role, Midwife Ira has prioritized improving the nutritional status of the Baduy community. Dompet Dhuafa supports this effort by distributing high-protein foods, such as eggs, known to enhance children’s growth in height and weight. These efforts target five villages within Kanekes Village, focusing on improving the health outcomes of children and mothers.

##### Targeted Nutritional Interventions for Stunting Prevention

At the time of the interview, Midwife Ira had served for seven months and identified stunting in over half of the children in her care. Among 63 children weighed across the five villages, over 40 were classified as stunted. To address this, she collaborates monthly with village midwives during Posyandu (integrated healthcare service post) visits. These sessions involve growth monitoring, nutritional assessments, and supplementary food distribution.

##### Challenges in Community Engagement and Attendance

Recognizing the challenges posed by the community’s demanding fieldwork, which limits attendance at Posyandu, Midwife Ira actively engages directly with the community.

“*For example, I want to give additional food like that, for example, giving eggs or doing an association for education and providing additional food like that, it turns out that the target is in the field, so there are only 1 or 2 people in the Village, it is terrible*”.(Midwife Ira, 2023)

##### Holistic and Comprehensive Stunting Interventions

Midwife Ira adopts a holistic approach to tackling stunting, emphasizing the importance of interventions across all life stages, from adolescence to early childhood. She provides education on stunting prevention and monitors nutritional status while distributing supplementary food. This comprehensive strategy acknowledges the need to address stunting through early education, particularly among young women and prospective mothers.

“*So indeed, stunting handling must be holistic and comprehensive. You cannot do it if it is just her mother. From before marriage, they also have to understand that*”.(Midwife Ira, 2023)

##### Plans: Investigating Nutritional and Parenting Practices

Midwife Ira plans to conduct detailed interviews with mothers to further her efforts to investigate underlying factors contributing to poor nutritional status, such as dietary habits and parenting practices. Mothers facing challenges will be invited to newly established nutrition posts to receive training on preparing supplementary foods and monitoring growth. The effectiveness of these interventions will be evaluated by reweighing children after 12 days.

#### 3.2.2. NGO Sahabat Relawan Indonesia (SRI)

##### SRI’s Pilot Projects to Address Stunting in Indonesia

SRI is an NGO that addresses Indonesia’s stunting and broader health challenges. The organization has established pilot projects in three regions: Kanekes Village (Baduy), the Ujung Kulon National Park vicinity, and Tunda Island. Within Kanekes Village, SRI actively monitors eight key locations across five southern villages, including Ciboleger, Cijangkar, and Cigula, to address stunting comprehensively.

“*Yes, there is. Our midwives currently focus on overcoming stunting and are affiliated with the NGO Dompet Dhuafa. So Dompet Dhuafa places one midwife to work at the same health centre, most of whom are sent here. Yes, we accept it. So we have monitoring at eight stunting points*”.(Chairman of SRI, 2023)

SRI’s approach extends beyond stunting, encompassing broader public health concerns, such as mental health. The organization has identified five individuals with mental health disorders within its operational areas. Additionally, SRI addresses various health challenges, including injuries from accidents like falls from trees. These efforts highlight SRI’s comprehensive commitment to improving the well-being of the Baduy community and other vulnerable populations under its care.

##### Stunting as a Key Public Health Concern in Kanekes Village

The chairman of SRI emphasized that stunting is a critical contributor to high child and maternal mortality rates in Kanekes Village. This issue originates in utero due to inadequate maternal nutritional intake, which adversely affects the growth and development of the fetus.

“*Stunting is not just a problem from birth. After working closely with doctors, they explained that stunting begins during pregnancy due to insufficient maternal nutrition. It prevents the mother from providing adequate nutrition to the fetus, adversely impacting growth and development from the womb*”.(Chairman of SRI, 2023)

##### Challenges in Maternal Nutrition and Health Documentation

Despite its severity, the lack of reporting and documentation on child and maternal deaths in health data complicates national intervention efforts and hinders the implementation of effective health policies.

##### Inadequacies in Sustainable Stunting Interventions

While significant funding has been allocated to stunting programs, the chairman noted that these initiatives often fail to provide sustainable solutions. Addressing stunting requires a continuous and consistent commitment, much like the treatment of chronic illnesses, such as tuberculosis. Interruptions in intervention schedules, such as delays in providing additional food to meet nutritional needs, can result in setbacks, necessitating the restart of interventions. It underscores the importance of maintaining uninterrupted efforts to achieve long-term success in combating stunting.

##### Leveraging Traditional Authority for Stunting Policy Advocacy

In Kanekes Village, traditional authority is most substantial in Baduy Dalam, presenting an opportunity to leverage a political approach to stunting intervention. The chairman highlighted that regional officials maintain familial ties with the Jaro Baduy Dalam (the traditional village head), which could be valuable in advocating for policy changes to reduce stunting prevalence. However, the success of such efforts requires robust government support. The health center must play a central role by facilitating treatment access, enabling midwives to conduct home visits, and improving the accessibility of ambulances for emergencies. Notably, many of the health center’s assets were initially established through the efforts of Midwife Rose, underlining the need for sustained institutional commitment.

##### Food Security and Nutritional Challenges in the Baduy Community

The Baduy community has a well-established food safety system, with strict prohibitions on selling rice outside the community. It ensures that all locally produced rice is reserved for internal consumption, promoting food security. However, the system’s simplicity limits its effectiveness in addressing comprehensive nutritional needs. Consequently, community members often rely on less nutritious alternatives, such as snacks and instant noodles, to diversify their diets. Processed snacks and artificially sweetened beverages frequently replace healthier traditional foods, such as *Gogodog Cau* (banana fritters), further exacerbating nutritional deficiencies.

##### Fatalistic Attitudes and Health Risks in Agrarian Lifestyles

Furthermore, a fatalistic attitude towards health issues is pervasive in the Baduy community, primarily due to the severe nature of many health problems and the unavailability of immediate solutions. Venomous snake bites, a significant health threat in this agrarian community, underscore these challenges. The lack of affordable and accessible antivenom exacerbates the problem, leaving the community without effective prevention or treatment options.

##### Culturally Aligned Interventions for Stunting Prevention

Addressing stunting in the Baduy community requires increasing public awareness of its long-term impact. The SRI informant noted that emphasizing cognitive deficits, such as impaired intelligence, is insufficient in a community with traditional restrictions on formal education. Instead, interventions should align with local customs and promote sustainable practices. For example, encouraging households to plant fruit trees provides a culturally acceptable solution that improves nutritional diversity and reduces the risk of stunting over time.

##### Barriers to Collaboration Between Traditional Leaders and Government Programs

Despite some promising opportunities for collaboration, several challenges remain. The SRI informant highlighted a genealogical connection between a district official and the Baduy traditional chief through marriage, which could facilitate the mobilization of the Baduy community for government programs. However, inadequate support to bridge the gap between traditional leaders and government initiatives undermines the potential for cohesive and impactful collaboration. Strengthening these connections is essential to ensure that stunting interventions are effectively implemented and widely accepted.

“*This is the son of this father, who married the son of Jaro Baduy Dalam. So, he is the same as Jaro. Structurally, his family is noble. He says A, yes, A, and everyone follows him. If he says B, it’s all B. It’s just that, again, you can’t do it without assistance*”.(Chairman of SRI, 2023)

##### Communication Gaps in Utilizing Health Assistance and Resources

The root of the problem lies in the passivity among health workers and the need for more clarity in the handover process of health assistance. Although midwives and health workers have secured formal approval from the Jaro for various aids, such as ambulances, there is a significant lapse in proactive communication. Neither the Jaro nor the aid providers effectively inform the community about the availability and utilization of these resources. For instance, despite legally approved ambulances by the Baduy community, accessing these services remains challenging due to inadequate public awareness and coordination.

“*One of the duties of the Ciboleger Community Health Center is to assist Baduy residents. It includes facilitating treatment and midwives to come to the houses, but it is not done. Including what was criticized by the Jaro. The existence of midwives used to be viral. It was only an icon in the quote of Baduy’s helper, and that’s why a lot of assistance came through her, including ambulances and all kinds, but it did not reach the Baduy community. The only regret is that it’s difficult for the Baduy to borrow an ambulance from her. So finally, they don’t have a health centre here*”.(Chairman of SRI, 2023)

The ambiguity of the handover process was revealed as a communication problem because there was no word for the public to use the existing facilities:

“*Later, try to talk to Jaro because all the assistance from donors everywhere must use Jaro’s signature. But, after the aid came down, there was nothing like, “Let us go ahead and use it”. Even though his facilities are complete. There is a 4 × 4 ambulance car, there is 1, and there are many facilities. I just knew that*”.(Chairman of SRI, 2023)

##### Midwife’s Competency and Training Limitations in Kanekes Village

One of the reasons why no one dares to let go of the community to use the facilities is that the midwives’ competence still needs to improve. Midwives still need to gain the skills to handle critical childbirth problems.

I asked*, “What is this?”*“*Upgrade the skills of midwives*”.“*Sir, we have not joined the APN*”.“*What is APN*?”“*Normal childbirth care. The goal is to reduce maternal and child mortality*”.(Chairman of SRI, 2023)

The institution has made efforts to enhance healthcare by funding training programs for midwives, aiming to boost their confidence in serving the Baduy community, particularly expectant mothers. This initiative was intended to facilitate ambulance services to transport patients from their homes to health centers or midwifery practices. However, despite these efforts, no midwives out of the ten planned for training participated over a year, resulting in no significant improvements in service delivery or the utilization of ambulance facilities.

“*Well, it turns out that after being given a scholarship for one midwife of 5 million, even though it was only 3.5 million for the ANC fee, it was only a week. Nevertheless, in terms of legality, the midwife cannot practice if she does not have the ANC certificate. It has been over a year here, and not one has participated out of the ten people we financed. That is not good. In the past, I asked to upgrade the midwives. He said it could also be done if no such thing existed. I said I could not. Even though it is allowed at the Puskesmas because they need it, legally, the formality must exist first. I said I was embarrassed if I wanted to ask again. Now, Midwife Rose is the head of Korbid and the Midwife Coordinator. Yes, he did not push there, either. Each health centre exists*”.(Chairman of SRI, 2023)

##### Potential of Health Communication to Improve Community Outcomes

The findings from the SRI informant indicate that addressing stunting in the Baduy community does not require extraordinary measures but rather a practical approach to communication and health literacy. The core issue lies in the lack of adequate health communication, which, if resolved, could significantly mitigate the problem. From a cultural standpoint, the Baduy community has demonstrated an openness to modern medical practices, driven by their recognition of their effectiveness in addressing various health concerns. This openness underscores the potential of health communication as a vital tool for improving outcomes within the community.

“*Yes, actually, awareness. The awareness that being healthy is pursued in various ways. So, Sardi, before coming to us, is it already the masseuse? Taken everywhere*”.(Chairman of SRI, 2023)

##### National-Level Debates on Indigenous Healthcare Access

Efforts to address public health challenges in the Baduy community have sparked debates at the national level. Despite receiving approval from the minister of health to facilitate an operation at RSCM (Cipto Mangunkusumo National Central General Hospital), the decision encountered resistance from several directorgenerals, who raised concerns about equity and resource prioritization. However, the chairman of SRI argued that the Baduy people, as an isolated Indigenous community located relatively close to Jakarta, deserve access to necessary medical care. This situation sheds light on a broader issue: if Indigenous communities near Jakarta struggle to access healthcare, the challenges for more isolated and distant Indigenous populations become even more pronounced.

“*Finally, the Minister said, “Okay, take it to RSCM”. The RSCM said, “Yes, sir, [this Baduy person] the important thing is to enter the hospital first. The administration can follow later”. It uses a special route if the Minister says specialization must be given. Yesterday, a Director General did not want to say, “Mr Arif, don’t be given more rights or special attention”. I said, “The problem is that they are isolated indigenous tribes that need attention and are close to Jakarta. It is possible to do it in Papua, Tenggarong, Kutai, or Nias. But what about those outsides if they are close to Jakarta [the capital]*”?(Chairman of SRI, 2023)

## 4. Discussion

This study highlights various aspects of the Baduy community’s local wisdom concerning health, including community-based practices, resistance to modern health interventions except during emergencies, nutritional intake management, strict customs surrounding technology use, the acceptance of fate, reliance on medicinal plants, waste management practices, customary rituals, and the rejection of formal identity. These findings reveal unique challenges and opportunities for health interventions within the Baduy community.

Unlike research on Indigenous groups such as the Aboriginal community in Australia, distinct contrasts emerge. For example, studies on Aboriginal communities often prioritize improving nutrition by reducing food prices and increasing food availability without encountering significant food taboos [26]. In contrast, the Baduy community faces complex challenges due to their persistent adherence to food restrictions. Unlike other Indigenous groups with food taboos limited to specific conditions, such as pregnancy or postpartum [27], the Baduy maintain such restrictions universally. These practices align with their simplicity philosophy, reflecting both their cultural values and economic realities.

The extreme poverty experienced by the Baduy further exacerbates their challenges. Poverty, coupled with cultural practices, limits health literacy development. For instance, food taboos and the necessity of extensive labor in the fields highlight their low income, forcing them to prioritize productivity over nutritional diversity. Addressing these challenges requires a tailored communication model that effectively conveys health information to the Baduy community. Such a model should encompass key components, including the communicator, audience, message, media, feedback mechanisms, and barriers to effective communication.

This study expands on prior research by exploring the intersection of geographic conditions, cultural practices, and health challenges within the Baduy community. Poor nutrition, influenced by their rugged terrain and reliance on local resources, is a critical factor contributing to stunting. These findings align with earlier studies highlighting limited access to and acceptance of health services among the Baduy [28]. However, this study adds depth by explicitly linking these factors to the community’s geographicisolation and dietary practices, offering a comprehensive perspective on their health challenges.

The finding that the Baduy Dalam community rejects modern health interventions except in emergencies corroborates the results of [6], who emphasized the importance of empathetic approaches and effective cross-cultural communication. However, this study demonstrates how this rejection is closely tied to their historical laws and strong customs, revealing deeper complexities in managing Baduy’s public health. Moreover, this study finds that the Baduy are very cautious in choosing food, rejecting anything not produced by themselves and prioritizing satiety over nutritional quality. It is consistent with [29], who documented specific prohibitions during pregnancy and childbirth within the Baduy community, reflecting their local beliefs in health management. This study broadens the scope by identifying how their food patterns, tied to tradition and religion, impact overall community health. While there is a rejection of modern medical interventions, the study also finds that some Baduy members recognize the need to adapt traditions to modern medical needs. It indicates the potential for more effective health communication if the appropriate approach is used. This finding aligns with [30], who emphasized the importance of involving traditional leaders in conflict resolution in Indigenous communities, which could be applied to health management.

The study also reveals a strong belief in fatalism within the Baduy community, especially regarding health issues like stunting and snake bites. This attitude is evident in their acceptance of their children’s physical conditions and their high-risk handling of snake bites. The belief that a child’s physical condition should not be compared to others and that snake bites are a fate to be accepted illustrates that fatalism plays a significant role in their health practices.

A previous study by [31] on the Chinese government’s communication strategies, which involved negative neutralization, positive reinforcement, and emotional mobilization to increase public trust, may serve as a reference for understanding why effective communication strategies are necessary to address fatalistic beliefs in the Baduy community. This study highlights the importance of structured communication approaches to change community behaviors, which aligns with [7], who found that communication interventions can reduce stunting through behavior change. Varni et al. [32] also showed that effective health communication can mediate the negative impact of illness on quality of life, justifying this study’s argument that proper health communication interventions in the Baduy community could be vital to reducing stunting. In this context, the fatalism observed in the Baduy could be addressed through more persuasive and targeted communication strategies, such as those proposed by [9], who emphasized the importance of religious leaders in preventing stunting. Furthermore, studies by [3] identified cultural factors contributing to stunting, such as restrictions on nutritious food and early marriage. These findings underscore the need to integrate cultural understanding into communication strategies, which can help formulate more effective, culturally sensitive approaches to addressing health issues in the Baduy community.

Findings related to medicinal plants and waste management indicate local solid wisdom but highlight challenges that must be addressed with approaches that respect tradition while introducing safer and more effective health practices. This study is consistent with [4], who noted the negative economic impact of stunting, emphasizing the importance of communication interventions that could help the Baduy overcome economic limitations and improve their access to better healthcare services.

The finding that health messages delivered by the community head (RT) tend to be general and emphasize the importance of maintaining health despite economic hardships is consistent with the concept of stunting as a complex issue influenced by various factors, including economic conditions. In this context, the RT’s approach reflects an awareness of the importance of health but also illustrates the limitations in access and resources for implementing ideal health practices. It supports theories stating that maternal anthropometry and economic conditions play crucial roles in preventing stunting [33].

How Baduy women do not fully comprehend more specific health messages about stunting, such as those conveyed through posters or oral explanations by midwives, highlights the need for cultural sensitivity in health communication. Health messages in cross-cultural contexts must be tailored to the local culture and needs. Adapting messages based on cultural context poses a primary challenge to achieving better health outcomes.

Cultural beliefs, such as the idea that stunting is primarily genetic, can significantly hinder the success of nutritional interventions. When communities believe stunting is an unchangeable hereditary condition, they may be less motivated to engage in preventive measures, such as improving maternal and child nutrition or adopting healthy feeding practices. This perception creates barriers to intervention efforts, as the underlying assumption dismisses the role of diet and healthcare in preventing or mitigating stunting. Public health programs must consider these misconceptions and address them through targeted educational campaigns emphasizing the importance of nutrition and environmental factors.

Addressing stunting in communities where cultural beliefs are deeply ingrained is challenging. These beliefs are often passed down through generations and may be tied to a community’s identity and tradition. Changing such perceptions requires a nuanced approach, as directly challenging long-held beliefs may be met with resistance or skepticism. Therefore, health interventions must be designed with a deep understanding of local values, ensuring that they respect cultural norms while gently introducing new ideas that align with scientific evidence [34].

Health programs that aim to reduce stunting must be tailored to the specific cultural context of the communities they serve. It involves providing information about proper nutrition and health practices and ensuring that this information is delivered in a way that resonates with local values and traditions. Programs can incorporate culturally significant foods and practices, demonstrating how traditional diets can be adapted to meet nutritional needs [35,36]. Engaging in community dialogues and using storytelling methods rooted in local culture can also enhance the effectiveness of health messages [37].

Community involvement is crucial for the success of any health intervention, particularly in areas with strong cultural traditions. Local leaders, such as religious figures, village elders, or other influential community members, can play a pivotal role in bridging the gap between public health messages and community beliefs. Through interpersonal communication, these leaders can help communicate the importance of nutrition and health interventions in ways more likely to be accepted by the broader population. Their endorsement can lend credibility to health programs, making it easier to change attitudes and behaviors over time [38,39,40,41].

Governments and policymakers must develop health policies that are flexible enough to account for cultural diversity. Policies should mandate the inclusion of cultural assessments in the planning and implementation of health programs, ensuring that interventions are relevant and respectful to the target population. Policies should also promote training healthcare workers in cultural competency, equipping them with the skills to address misconceptions and foster trust within their communities. Improvements in educational and policy interventions were needed to support healthy feeding practices for infants and young children [42,43,44].

Non-governmental organizations (NGOs) and government agencies working in a community with a specific cultural context should adopt a community-based approach. It includes collaborating with local leaders, conducting cultural sensitivity training for staff, and developing health messages that align with the community’s values. NGOs and agencies should also focus on long-term engagement over short-term interventions, as changing cultural beliefs is a long, arduous process that requires continuous effort and trustbuilding.

## 5. Conclusions

This study explores the perceptions of stunting within the Baduy community and their implications for accepting health interventions. A key finding is the community’s predominant attribution of stunting to genetic factors. This belief is deeply rooted in their cultural worldview, which often links health outcomes to hereditary and spiritual influences. While there is some awareness of the role of nutrition, the prevailing genetic explanation tends to overshadow evidence-based approaches, particularly those emphasizing dietary strategies to combat stunting. This cultural perspective presents a significant challenge to implementing effective health interventions.

Addressing this issue requires a balanced approach that respects traditional beliefs while promoting the benefits of nutritional strategies. Future research should prioritize longitudinal studies to evaluate the effectiveness of culturally sensitive interventions. Such studies could provide critical insights into tailoring health programs that align with the community’s cultural values while encouraging evidence-based practices. Additionally, examining the long-term sustainability of dietary and health behavior changes will be crucial to understanding how such interventions can be maintained within the Baduy community.

Further research should also investigate innovative methods for engaging the community to enhance the success of health interventions. For example, assessing the influence of local leaders and community influencers in driving behavioral changes could significantly improve the acceptance of nutritional programs. Evaluating the impact of community-based approaches will provide a deeper understanding of addressing stunting effectively in culturally diverse settings. By integrating these insights, researchers and policymakers can develop strategies that address critical health challenges while respecting and incorporating the cultural values of Indigenous communities like the Baduy.

## Figures and Tables

**Figure 1 ijerph-22-00145-f001:**
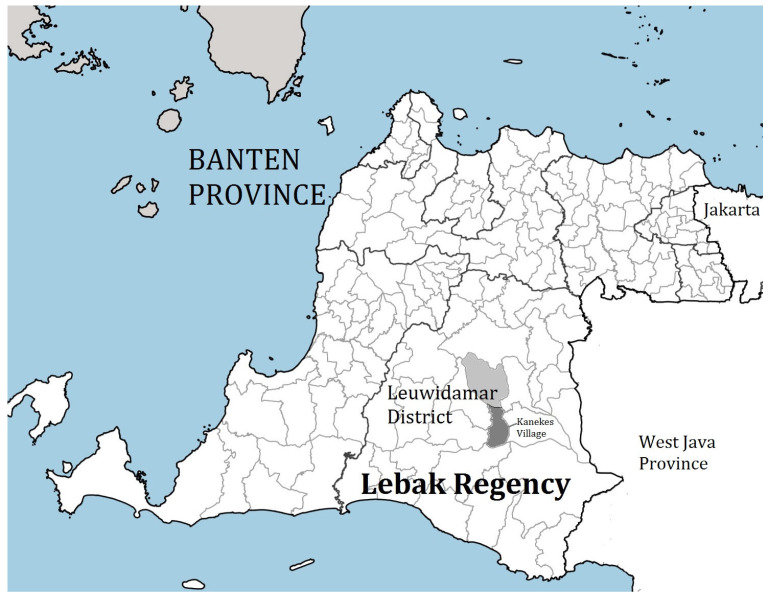
Location of the research in the Kanekes Village, Leuwidamar district, Lebak Regency, Banten Province, Indonesia [20].

**Figure 2 ijerph-22-00145-f002:**
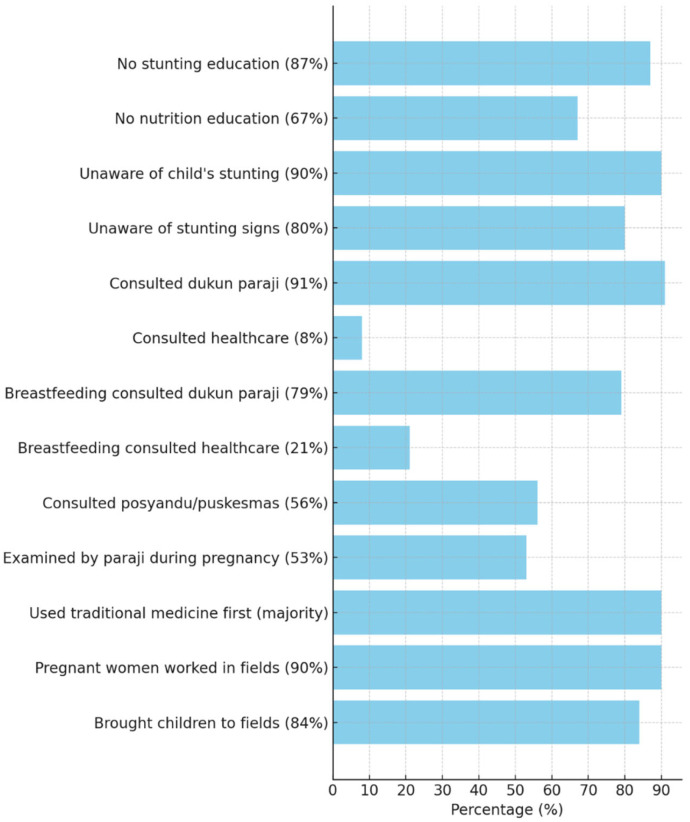
The survey results of 77 housewives in 10 Baduy villages [24].

**Table 1 ijerph-22-00145-t001:** Health problems in Baduy Village in 2022 and 2023.

Indicator	2022	2023
Number of babies/toddlers monitored nutritional status (fostered)	46	69
Number of undernourished infants/toddlers (Weight/Age)	10	69
Number of babies/toddlers short/very short/stunted (Height/Age)	20	31
Number of malnourished infants/toddlers (Weight/Age) and short/very short/stunting (Height/Age)	30	48
The number of undernourished infants/toddlers has improved	6	6
The number of babies/toddlers/very short/stunted has improved.	4	4
The number of malnourished and short/very short/stunted babies/toddlers has improved.	4	4

Source: Midwife Ira, 2023.

## Data Availability

The qualitative data used to support the findings of this study are available from the corresponding author upon reasonable request. The data are not publicly available due to privacy and ethical restrictions, as they contain information that could compromise the confidentiality of research participants.

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
