# Peer review of "Genetic Perception Versus Nutritional Factors: Analyzing the Indigenous Baduy Community’s Understanding of Stunting as a Health Issue"

_ijerph, 2025, doi:10.3390/ijerph22020145_

Round 1

Reviewer 1 Report

Comments and Suggestions for Authors

This paper highlights a significant issue of interest: how culture influences health outcomes and the need for culturally safe interventions/practices to influence behavioural change and, ultimately, health outcomes. It also highlights the importance of integrating Indigenous knowledge and practices to inform health behaviours and practices.

Methods - the authors need to describe the positionality of the researchers/s as this impacts how the questions were asked and interpreted - very important when conducting this type of research.

The authors present a rich content of evidence. However, under the results section, under each sub-section, e.g. 3.1, for clarity, more subheadings to highlight specific themes under the broader theme are needed, as this helps with the flow of the paper. 

Comments on the Quality of English Language

There are some grammatical errors and the authors need to revise their intext referencing e.g. pg2 "The consequences of stunting are severe, including impaired cognitive development, increased susceptibility to diseases, and reduced future earning potential, as highlighted by (4), who estimate that stunting could reduce individual earnings by up to 7%". The bolded section - proper intext referencing required. 

Author Response

Thank you so much for the reviews. We add positionality statement in the methods section:

“In this study, the researcher's position is as an outsider positionality (Bukamal, 2022). Researchers can be said to be outsiders because they have cultural differences (Yip, 2024) with the Baduy community. Researchers are academics who live in an urban environment with a modern culture and academic mindset while the Baduy community lives in a rural environment with their own culture. As outsiders, we are less likely to form biases because they are not embedded in the social or cultural context of the group being studied; have emotional distance that reduces the likelihood of personal involvement influencing the research outcomes; bring a fresh, external viewpoint that might uncover nuances or issues that insiders might overlook due to familiarity; can focus on evidence-based conclusions rather than relying on ingrained beliefs or norms; and able to critically analyze the context and question practices or beliefs that insiders might take for granted (Kham, 2024). However, this position restricted our ability to fully access the complex social networks, cultural subtleties, and localized knowledge (Bandauko, 2024) that define the Baduy community. However, this status as an outsider is not a fixed entity (Dahal, 2023) because over time, we have become more accepted by the Baduy community and more involved in their simple cultural context.”

Furthermore, now we put subheadings in the results section to highlight specific themes and giving clarity to the flow of the paper. Grammatical errors corrected and referencing style followed MDPI style.

Reviewer 2 Report

Comments and Suggestions for Authors

I found few typo errors in the article, kindly read it carefully and correct it.

Described Methodology is self-made or followed from the previous study, no one references were cited, and it shows the unscientific and vague way of approach for the study which author carried out.

Cited references in the result part were different in format compared to introduction and discussion.

Actual data not mentioned in the results, which is evident for the current findings and also it more valid to justify the work and discuss appropriately.

Authors need to read whole article few times, and many paragraphs should be re written and flow should come properly and scientifically.

Work is excellent, it is acceptable but mentioned comments should be considered seriously and answer it or incorporate it in all the sections to improve the article.

Comments on the Quality of English Language

Authors need to read whole article few times, and many paragraphs should be re written and flow should come properly and scientifically.

Author Response

Thank you for the review. We read the article several times and corrected typo errors in the article. We also change the referencing format to MDPI style using Mendeley Reference Manager to make the result part the same format as other parts of the paper. Around 20 paragraphs were rewritten to make them come properly and scientifically.

Eleven new references were added to the methodology section. Informants and data analysis method clarified. Anyway, what do you mean by actual data? the statistics we presented in the results section is secondary data, not our work.

Furthermore, now we put subheadings in the results section to highlight specific themes and giving clarity to the flow of the paper. Grammatical errors were corrected and referencing style followed MDPI style.
